# Heterospin frustration in a metal-fullerene-bonded semiconductive antiferromagnet

Yongbing Shen [1 ✉], Mengxing Cui[1], Shinya Takaishi [1], Hideyuki Kawasoko [1], Kunihisa Sugimoto [2], Takao Tsumuraya [3], Akihiro Otsuka [4,5], Eunsang Kwon[6], Takefumi Yoshida[1], Norihisa Hoshino [7], Kazuhiko Kawachi[8], Yasuhiko Kasama[8], Tomoyuki Akutagawa[7], Tomoteru Fukumura[1,9] & Masahiro Yamashita[1,10 ✉]

Lithium-ion-encapsulated fullerenes ($Li^+@C_{60}$) are 3D superatoms with rich oxidative states. Here we show a conductive and magnetically frustrated metal–fullerene-bonded framework $\{[Cu_4(Li@C_{60})(L)(py)_4](NTf_2)(hexane)\}_n$ (**1**) ($L = 1,2,4,5$-tetrakis(methanesulfonamido) benzene, py = pyridine, $NTf_2^- =$ bis(trifluoromethane)sulfonamide anion) prepared from redox-active dinuclear metal complex $Cu_2(L)(py)_4$ and lithium-ion-encapsulated fullerene salt $(Li^+@C_{60})(NTf_2^-)$. Electron donor $Cu_2(L)(py)_2$ bonds to acceptor $Li^+@C_{60}$ via eight Cu–C bonds. Cu–C bond formation stems from spontaneous charge transfer (CT) between $Cu_2(L)(py)_4$ and $(Li^+@C_{60})(NTf_2^-)$ by removing the two-terminal py molecules, yielding triplet ground state $[Cu_2(L)(py)_2]^+(Li^+@C_{60}^{\bullet-})$, evidenced by absorption and electron paramagnetic resonance (EPR) spectra, magnetic properties and quantum chemical calculations. Moreover, $Li^+@C_{60}^{\bullet-}$ radicals ($S = ½$) and $Cu^{2+}$ ions ($S = ½$) interact antiferromagnetically in triangular spin lattices in the absence of long-range magnetic ordering to 1.8 K. The low-temperature heat capacity indicated that compound **1** is a potential candidate for an $S = ½$ quantum spin liquid (QSL).

[1] Department of Chemistry, Graduate School of Science, Tohoku University, 6-3 Aza-Aoba, Aramaki, Sendai 980-8578, Japan. [2] Diffraction & Scattering Division Synchrotron Radiation Research Institute, Hyogo 679-5198, Japan. [3] Priority Organization for Innovation and Excellence, Kumamoto University, 2-39-1 Kurokami, Kumamoto 860-8555, Japan. [4] Division of Chemistry, Graduate School of Science, Kyoto University, Sakyo-Ku, Kyoto 606-8502, Japan. [5] Research Center for Low Temperature and Materials Sciences, Kyoto University, Sakyo-Ku, Kyoto 606-8501, Japan. [6] Research and Analytical Center for Giant Molecules, Tohoku University, 6-3, Aramaki-Aza-Aoba, Aoba-ku, Sendai 980-8578, Japan. [7] Institute of Multidisciplinary Research for Advanced Materials, Tohoku University, 2-1-1 Katahira, Aoba-Ku, Sendai 980-8577, Japan. [8] Idea International Co., Ltd., 1-15-35 Sagizamori, Aoba-ku, Sendai 981-0922, Japan. [9] Advanced Institute for Materials Research and Core Research Cluster, Tohoku University, Sendai 980-8577, Japan. [10] School of Materials Science and Engineering, Nankai University, Tianjin 300350, China. ✉email: shen.yongbing.b1@tohoku.ac.jp; yamasita@agnus.chem.tohoku.ac.jp

Since the pure form of the lithium-ion-encapsulated fullerene salt [Li@$C_{60}$](SbCl$_6$) was isolated and structurally determined by X-ray diffraction analysis in 2010[1], studies on this smallest endohedral metallofullerene (EMF) have taken precedence over photoinduced electron transfer (ET) in non-covalent donor–acceptor (D–A) complexes[2–6], covalent metal complexes[7], organic photovoltaics[8], and molecular electronics[9] over the past decade because of the salt's unique structure and electronic properties relative to pristine $C_{60}$[10–14]. Although the optical bandgap ($E_g$) is very close to that of pristine $C_{60}$, the lowest unoccupied molecular orbital (LUMO) of Li$^+$@$C_{60}$ has been observed to decline significantly to −3.90 eV with an initial reduction potential at −0.39 V versus Fc/Fc$^+$ in $o$-dichlorobenzene ($o$-DCB)[1]. Furthermore, the oxidisation state of Li$^+$@$C_{60}$ can be easily tuned from 1+ to 3− by external chemical stimuli to realise various electronic states[1]. Therefore, this compound has been widely used as a π-electron acceptor owing to the small reorganisation energy required, which leads to highly delocalised π-electrons over a 3D sphere[2,3]. It is possible that Li$^+$@$C_{60}$ can be doped by alkali metals to produce A$_3$(Li@$C_{60}$) species (A = K$^+$, Rb$^+$, and Cs$^+$) in imitation of A$_3$C$_{60}$ superconductors with three electrons accommodated in the triply degenerated LUMO[15–18]. In particular, the emergence of Li$^+$@$C_{60}$$^{•−}$ requires a milder chemical oxidant than for C$_{60}$$^{•−}$. Such a low reduction potential provides many opportunities for coordination chemistry. According to the Mulliken theory[19–22], the formation of the CT complex requires efficient orbital overlap between the highest occupied molecular orbital (HOMO) of D and the LUMO of A. The CT interactions in the ground state increase with a decrease in the energy difference between the HOMO of D and LUMO of A. In this regard, Martin and co-workers reported several CT complexes between comparatively strong electron donors such as π-extended tetrathiafulvalene derivatives and fullerenes[23,24]. Additionally, Yamada et al. observed a triplet charge-separated state by laser-exciting a curved π-surface donor and Li$^+$@$C_{60}$ in solution[3].

Although $C_{60}$ is an ideal ligand to realise topological architectures owing to its isotropic coordination environment[25], to date, reports on the preparation of electrically conductive and magnetically frustrated solids based on Li$^+$@C60 remain elusive. On the other hand, the $S = ½$ electronic system holds promise for exploring interesting quantum phenomena such as unconventional superconductivity[26–29] and QSLs, which are commonly observed in triangular or kagomé lattices[30–36]. Herein, we selected an electronically active donor, Cu$_2$(L)(py)$_4$, and a 3D charge-tuneable metallofullerene, Li$^+$@$C_{60}$, as both the ligand and electron acceptor to construct a conductive and spin-frustrated framework based on the Mulliken theory.

In this work, an $S = ½$ electronic framework {[Cu$_4$(Li$^+$@$C_{60}$)(L)(py)$_4$](NTf$_2$)(hexane)}$_n$ (**1**) is isolated by constructing the donor and acceptor. The donor is bonded to the acceptor via Cu–C bonds. It is noteworthy that the HOMO of Cu$_2$(L)(py)$_4$ has the equivalent energy level of the LUMO of Li$^+$@$C_{60}$, thereby facilitating CT interactions in the ground state. From our calculations, we find that the $d_{xz}$ orbitals of the Cu ions in Cu$_2$(L)(py)$_4$ are delocalised and strongly coupled with the N($p_z$) orbitals (π-electrons) of the ligand, thereby yielding delocalised electrons in the HOMO. In **1**, the four-terminal $d_{xz}$ orbitals of Cu ions coordinate with one Li$^+$@$C_{60}$ cage and the remaining four Cu ions coordinate with the next Li$^+$@$C_{60}$ cage to form an infinitely scaled 1D chain structure. The four Cu(L)(py) sites transfer a single electron into the Li$^+$@$C_{60}$ cage, and the resulting four Cu$^{2+}$ ($S = ½$) ions and Li$^+$@$C_{60}$$^{•−}$ ($S = ½$) interact with each other and exhibit magnetic frustration in a triangular-like lattice. Our study demonstrates long-range electrical conductivity ($\sigma$) and spin frustration using Li$^+$@$C_{60}$ superatoms in such a bonded D–A-type framework.

## Results and discussion

**Metal-fullerene bonded donor–acceptor-type framework.** Compound **1** was synthesised by reaction of Cu$_2$(L)(py)$_4$ and (Li$^+$@$C_{60}$)(NTf$_2$$^−$) with a molar ratio of 2:1 in $o$-DCB solution. It crystallises in the triclinic $P\bar{1}$ space group with the unit-cell dimensionality of $a = 9.9963(3)$ Å, $b = 13.3087(3)$ Å, $c = 19.7031(5)$ Å, $\alpha = 77.323(2)°$, $\beta = 76.572(2)°$, and $\gamma = 69.500(2)°$ at 120 K. One Li$^+$@$C_{60}$ cage coordinates with four Cu ions from four Cu$_2$(L)(py)$_2$ molecules (Fig. 1a), and the remaining four Cu ions coordinate with the next Li$^+$@$C_{60}$ cage to afford infinite 1D ladder-like structures along the $b$-axis (Fig. 2a). Each Cu ion exhibits an equivalent of five coordination numbers (two Cu–C and three Cu–N bonds) in a distorted trigonal bipyramid geometry (Supporting Information Fig. 1). The rectangular plane formed by the four Cu ions perfectly divides the $C_{60}$ cage into two (Fig. 1b), and the shell-like Li$^+$ ion is off-centred and localised above or below the Cu plane with thermal ellipsoids at 50% probability. Similar Li$^+$-ion arrangements are also observed in the other forms of Li$^+$@$C_{60}$ salts[37,38]. The inner Li$^+$ ion coordinates with the six-carbon ring with Li–C bond lengths in the range of 2.337–2.527 Å, which are consistent with the reported results[7,39]. The encapsulated Li$^+$ ions should strengthen the π back-bonding from the transition-metal centre to the fullerene cage[7]; however, the Cu ions do not coordinate with the rings of the six-carbon (pink atoms in Fig. 2) owing to the highly symmetrical array of the four Cu ions.

The benzene ring of the ligand is slightly distorted (Supporting Information Fig. 2), and the bond patterns indicate the loss of aromaticity (Fig. 1c). On the other hand, the hexane molecules are trapped inside the squared structure of [Cu$_2$(L)(py)$_3$(Li$^+$@$C_{60}$)]$_2$ and strongly disordered along two lines (Supporting Information Fig. 3). Close to the square structure, the NTf$_2$$^−$ molecules interact

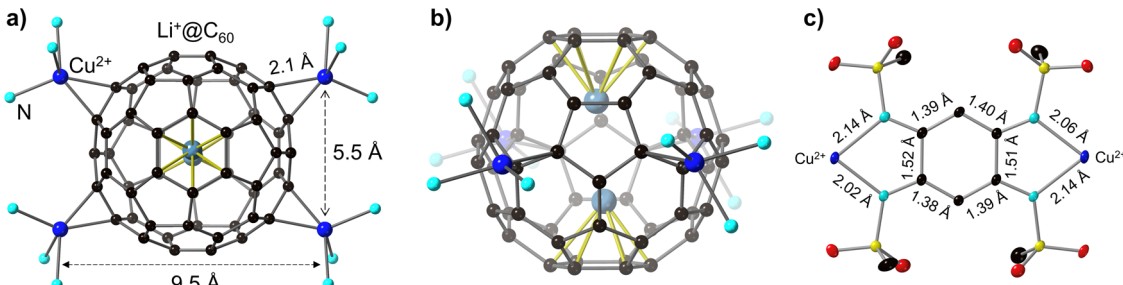

**Fig. 1 Crystal structures of Cu$_4$-Li$^+$@$C_{60}$ and Cu$_2$(L).** **a** The coordination geometry of [Cu(N)$_3$]$_4$(Li$^+$@$C_{60}$) and the arrangement of Li-ion. One Li$^+$@$C_{60}$ coordinates with four Cu ions. The four Cu ions form a rectangular-like plane with a length and width of 9.5 and 5.5 Å, respectively. **b** The Li-ion coordinates with six carbon-ring and locates above or below the Cu-plane with thermal ellipsoids at the 50% probability. The Cu-plane perfectly splits the Li$^+$@$C_{60}$ cage in half. **c** The structure of the donor, Cu$_2$(L) and its bond length. Colour code: C (black), Cu (blue), N (cyan), S (yellow), O (red) and Li (pastel blue).

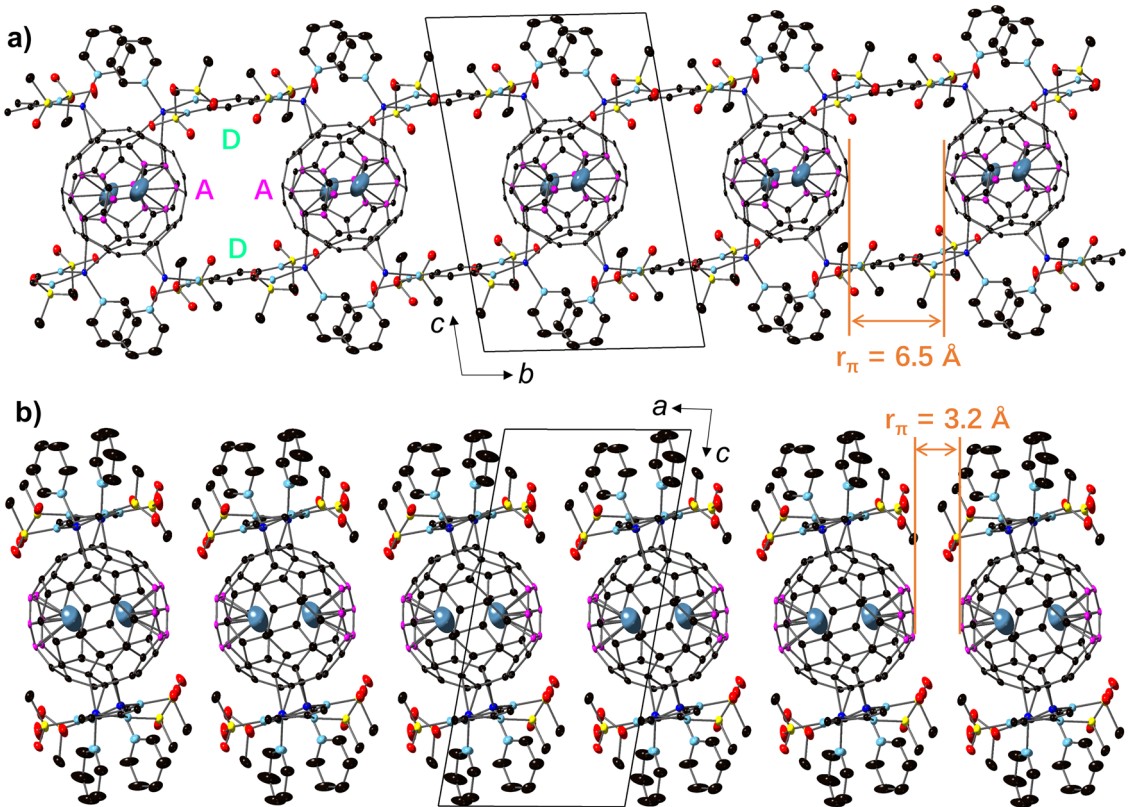

**Fig. 2 The structure arrangements of the ladder-like chains. a** 1D ladder-like structure along the *b*-axis in the *bc* plane (anion NTf$_2^-$ and hexane molecules are omitted). One Li$^+$@C$_{60}$ cage coordinates with four Cu ions and the remaining four-terminal Cu ions coordinate with the adjacent Li$^+$@C$_{60}$ cage to form a 1D ladder-like structure. Distance $r_\pi$ between adjacent Li$^+$@C$_{60}$ cages is 6.5 Å. The Li atoms are off-centred and polarised above/below the Cu plane. **b** Top view of the 1D ladder-like structure in the *ac* plane. Each structure stacks along the *a*-axis to form a 2D interacting sheet with intermolecular distance = 3.2 Å between two Li$^+$@C$_{60}$ cages. Colour code: Cu (blue), sulphur (yellow), oxygen (red), nitrogen (pastel cyan), carbon (black/pink). Hydrogen atoms are omitted for clarity.

electrostatically with ligands through hydrogen bonds. Inside the ladder chain, the π···π distance ($r_\pi$) between the two Li$^+$@C$_{60}$ cages is 6.5 Å (Fig. 2a), indicating well-separated π electrons along the *b*-axis. The 1D ladder structures self-stack along the *a*-axis to form 2D interacting nanosheets with $r_\pi$ of 3.2 Å (Fig. 2b). To determine the structure phase purity, the temperature dependence of crystal structures were determined at various temperature points (25, 50, 100, 200, and 300 K) by using synchrotron radiation. The structure analysis indicated that there are no structural changes or distortions in 25–300 K (PXRD patterns in Supporting Information Fig. 4). Above 100 K, we did not detect the Li$^+$ position because of the fast motion of the Li$^+$ ion inside the cage. Below 100 K, the Li$^+$ ion was disordered and localised at two equivalent positions due to the symmetry. As the localised Li$^+$ ions strongly attract the negative radicals (via Li–C bonds), the C$_{60}^{•-}$ radicals localised on the six-carbon ring (pink atoms in Fig. 2).

**Spontaneous charge transfer between Cu$_2$(L)(py)$_4$ and (Li$^+$@C$_{60}$) (NTf$_2^-$) by precise control of the redox activities.** Figure 3a shows the CV plot to Fc/Fc$^+$. Four reversible redox processes with the first and second oxidisation potentials are observed at −0.36 and −0.04 V, respectively. Meanwhile, the first and second reduction potentials are observed at −0.92 and −1.08 V, respectively. The potential at −0.36 V corresponds to the first oxidisation process of Cu$_2$(L)(py)$_4$ to [Cu$_2$(L)(py)$_4$]$^+$. In this regard, Aoyagi et al. reported that the first reduction process from Li$^+$@C$_{60}$ to Li$^+$@C$_{60}^{•-}$ occurs at −0.39 eV in *o*-DCB versus Fc/Fc$^{+1}$. Consequently, the HOMO energy level of Cu$_2$(L)(py)$_4$ can be simply treated as equivalent to the LUMO energy level of Li$^+$@C$_{60}$ if the

dissolution-free energy effect is ignored (Fig. 3b). According to the Mulliken theory, ET from Cu$_2$(L)(py)$_4$ (donor) to Li$^+$@C$_{60}$ (acceptor) can spontaneously occur without external energy to generate a triplet ground state [Cu$_2$(L)(py)$_4$]$^+$[Li$^+$@C$_{60}^{•-}$]. The solution-state absorption spectra in Fig. 3c evidence this mechanism. (Li$^+$@C$_{60}$)(NTf$_2^-$) does not show any absorption band from 750 to 1500 nm whereas Cu$_2$(L)(py)$_4$ shows a strong and broadband at 920 nm. Once the two pristine molecules are mixed in *o*-DCB, two new bands at 886 and 1032 nm are observed, strongly indicating that Li$^+$@C$_{60}^{•-}$ is generated[40]. Moreover, the strong band at 920 nm in Cu$_2$(L)(py)$_4$ vanishes from the mixed solution, strongly indicating ET occurrence. The solid-state absorption spectrum of **1** shows several broad absorption bands at 1.30, 1.91, 2.42, and 3.48 eV (Supporting Information Fig. 5), where the band at 1.30 eV extends to the IR region (inset of Supporting Information Fig. 5), which indicates that **1** has a small optical bandgap ($E_g$). Figure 3d shows the Tauc plot of the Kubelka–Munk-transformed spectrum, with $E_g$ = 0.57 eV, as obtained from a linear fit to the low-energy onset of absorption. However, it is difficult to assign the Li$^+$@C$_{60}^{•-}$ band owing to absorption-band superpositions. Nevertheless, the generated Li$^+$@C$_{60}^{•-}$ can be detected by the EPR spectra. Figure 3e shows the temperature dependence of the EPR spectra of **1** for 3.5–300 K with all spectra showing two EPR active bands. At the lower magnetic field, parallel $g_{II}$ values of 2.45(3), 2.32(1), and perpendicular $g_\perp$ value of 2.09(1) are observed at 300 K. The *g* values and peak-to-peak linewidth, $\Delta H_{PP}$ = 217.5(4) G, are entirely consistent with the Cu$^{2+}$ ion ($S = ½$)[41,42]. Another weak signal observed near the Cu$^{2+}$ signal with $g = 2.0008(3)$ and $\Delta H_{PP}$ =

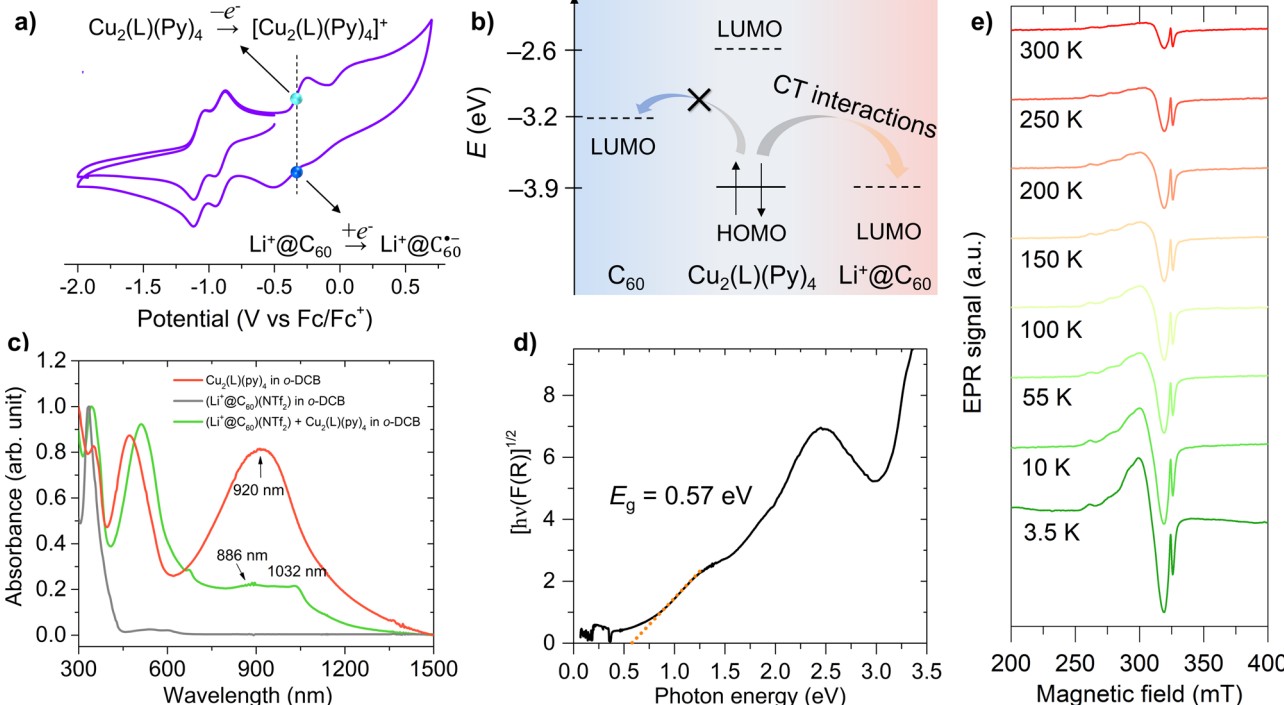

**Fig. 3 Spectroscopic characterization. a** Cyclic voltammogram (−2.0 to 0.8 V versus Fc/Fc⁺) of $Cu_2(L)(py)_4$ in o-DCB with 0.1 M TBAPF₆ as the supporting electrolyte. The pale blue dot at $E = -0.39$ eV represents the first reduction potential from $Li^+@C_{60}$ to $Li^+@C_{60}^{\bullet-}$. **b** Schematic of charge-transfer interactions of HOMO and LUMO orbital energies between $Cu_2(L)(py)_4$ and $Li^+@C_{60}$ calculated from the cyclic voltammogram. The HOMO energy level of $Cu_2(L)(py)_4$ is identical to the LUMO energy level of $Li^+@C_{60}$, indicating strong CT interactions. **c** In situ solution-state UV–Vis–NIR absorption spectra at room temperature. **d** Tauc plot of room-temperature diffuse reflectance UV–vis–NIR spectra of **1** from 0.05 to 3.3 eV, obtained via Kubelka–Munk transforms (F(R)) for indirect allowed transition. The optical bandgap is determined as 0.57 eV by the linear fit (orange dots) of absorption onset in the NIR energy region. **e** First-derivative solid-state X-band absorption EPR spectra of **1** for 3.5–300 K.

8.1(2) G, signifies the presence of electronically active $Li^+@C_{60}^{\bullet-}$ radical[3,40,43]. However, it is difficult to assign significant interactions between $Cu^{2+}$ ions and $Li^+@C_{60}^{\bullet-}$ radical from the EPR spectra, probably owing to the superposition of the two characterised bands, but the spectra demonstrate that spontaneous ET occurs from $Cu_2L(py)_4$ to $Li^+@C_{60}$ and **1** remains in a triplet ground state up to 3.5 K.

***S = ½ heterospin frustration between $Cu^{2+}$ and $Li^+@C_{60}^{\bullet-}$ in the ladder-like chains.*** The intensity of main peaks in the EPR spectra mimics the thermal dependence of the magnetic susceptibility ($\chi$). The $\chi$−T plot of **1** does not show any magnetic phase transition from 1.8 to 300 K under 1 T (Fig. 4a) and 4 T fields (Supporting Information Fig. 6), thereby indicating no long-range magnetic ordering. In addition, no sign of a spin-glass state is observed in the rectangular Cu lattices, as there is no deviation between the zero-field-cooled (ZFC) and field-cooled (FC) measurements. This suggests that $Li^+@C_{60}^{\bullet-}$ radicals are involved in the magnetic reactions. Fitting the $(\chi - \chi_0)^{-1}$ to the Curie–Weiss law at high temperatures ($T > 175$ K) yields a large negative Curie–Weiss temperature, $\theta_{cw} = -190$ K, suggesting strong antiferromagnetic (AF) interactions between the spins ($\chi_0$ is defined as core diamagnetic or Van–Vleck paramagnetic susceptibility[44] and determined to be $8.8 \times 10^{-4}$ cm³ mol⁻¹ in Supporting Information Fig. 7). Even the fitted low-temperature $\chi^{-1}$ ($T = 1.8–10$ K) results in a $\theta_{cw} = -1.6$ K, indicating significant AF interactions. The $(\chi - \chi_0)T$ value at 300 K is 1.71 cm³ K mol⁻¹ (Supporting Information Fig. 8), which follows the theoretical prediction of four $Cu^{2+}$ ions and one $Li^+@C_{60}^{\bullet-}$ radical per unit ($\chi T_{calc} = 0.375 \times 5 = 1.875$ cm³ K mol⁻¹). The rapid decrease in $(\chi - \chi_0)T$ with temperature reduction indicates that AF exchange interactions are

dominant. In addition, the magnetic-field dependence of magnetisation from 1.8 to 300 K does not show any visible hysteresis loops (Fig. 4b).

To elucidate such strong exchange coupling in **1**, we consider Fig. 4c, which shows the spin arrangements in a ladder-like structure with four $Cu^{2+}$ ions positioned at the corner of the rectangular plane and one $Li^+@C_{60}^{\bullet-}$ at the centre. In this magnetic pattern, owing to symmetry, one $Cu^{2+}$ ion magnetically interacts with three adjacent $Cu^{2+}$ ions ($J_{12}$ and $J_{13}$), which are magnetically linked by the $Li^+@C_{60}^{\bullet-}$ radicals. Moreover, the $Cu^{2+}$ ions interact with $Li^+@C_{60}^{-}$($J_{Cu-C_{60}^{-}}$) via Cu–C bonds. Such complicated and competitive interactions lead to a magnetically frustrated ground state. To obtain the J values, we used the following spin Hamiltonian $\hat{H}$ using Eq. 1 by considering two kinds of exchange coupling $J_{Cu-Cu}$ and $J_{Cu-C_{60}^{-}}$.

$$
\begin{aligned}
\hat{H} = &-2J_{Cu-Cu}(\hat{S}_{Cu1} \cdot \hat{S}_{Cu2} + \hat{S}_{Cu3} \cdot \hat{S}_{Cu4}) \\
&-2J_{Cu-C_{60}^-}(\hat{S}_{Cu1} \cdot \hat{S}_{C_{60}^-} + \hat{S}_{Cu2} \cdot \hat{S}_{C_{60}^-} + \hat{S}_{Cu3} \cdot \hat{S}_{C_{60}^-} + \hat{S}_{Cu4} \cdot \hat{S}_{C_{60}^-}) \\
&+ \mu_B \bar{B} \cdot g_{Cu} \cdot \hat{S}_{Cu} + \mu_B \cdot g_{C_{60}^-} \cdot \bar{B} \cdot \hat{S}_{C_{60}^-}
\end{aligned} \quad (1)
$$

To avoid overparameterization, we assumed the interactions of $J_{14}$ and $J_{23}$ are ignored compared to $J_{12}$, $J_{34}$ and $J_{Cu-C_{60}^{-}}$ due to their long metal-metal distances and the interactions between $C_{60}^{\bullet-}$ and Cu1, Cu2 are equivalent. The effective exchange coupling parameters of $J_{Cu-Cu} = -170(5)$ K and $J_{1-C_{60}^{-}} = J_{2-C_{60}^{-}} = -185(3)$ K with $g_{Cu} = 2.09(0)$ were obtained after the best fit ($g_{C_{60}^{\bullet-}}$ was fixed to 2.0). The negative $J_{Cu-Cu}$ and $J_{Cu-C_{60}^{-}}$ values confirm that any two neighbouring spins are AF coupled. Therefore, in this rectangular spin-lattice, composed of several triangular lattices (such as $\Delta Cu_1Cu_2C_{60}^{\bullet-}$), we can observe the three spins exhibiting competing interactions. Frustration

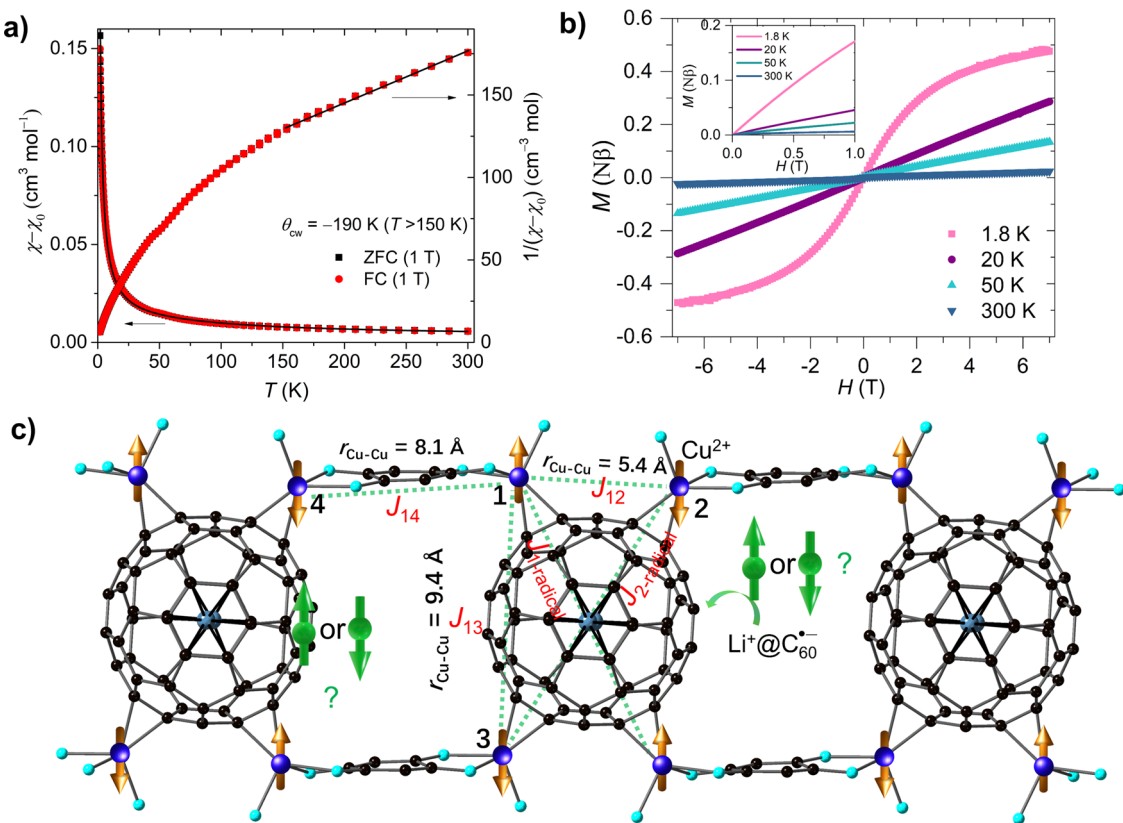

**Fig. 4 Magnetic properties. a** The temperature dependence of magnetic susceptibility product ($\chi - \chi_0$) of **1** in FC, ZFC modes and the corresponding ($\chi - \chi_0$)$^{-1}$–$T$ plots under a 1 T field from 1.8 to 300 K. The black curve represents the best fit for ($\chi - \chi_0$) by considering the possible exchange interaction between spins. The black line represents the best fit by the Curie–Weiss law. A large negative $\theta_{cw} = -190$ K is observed, suggesting strong AF exchange interactions. **b** Magnetic-field dependence of magnetisation of **1** at 1.8, 20, 50, and 300 K. There is no hysteresis loop at these temperatures. **c** Spin orientations of the Cu$^{2+}$ ions and Li$^+$@C$_{60}{}^{\bullet-}$ superatoms emerging in the 1D ladder-like magnetic chain; the four Cu ions and Li$^+$@C$_{60}{}^{\bullet-}$ superatoms are aligned in a triangular-like lattice and antiferromagnetically interact with each other.

parameter[45] $f = |\theta_{cw}| / T_N$ is >105 ($T_N < 1.8$ K), indicating that **1** has a highly frustrated ground state. In addition, the real-part alternating current magnetic susceptibility ($\chi'$) measurements indicated the absence of magnetic ordering to 1.8 K (Supporting Information Fig. 9). Such a frustrated lattice leads to inner Li$^+$ ions locating far away from the four coordinated Cu$^{2+}$ ions. Moreover, because of the pull forces from the eight external Cu–C bonds and six internal Li–C bonds, the Li$^+$@C$_{60}$ cage is geometrically distorted with the diagonal lengths of 6.4 and 7.1 Å, respectively (Supporting Information Fig. 10).

**Heat capacity and electrical conductivity.** To further demonstrate the absence of magnetic phase transitions, the temperature dependence of heat capacity ($C$) was measured. The $C$ versus $T$ and $C/T$ versus $T$ plots (Fig. 5a, b) clearly showed that there is no sharp thermal anomaly in 2–20 K, indicative of no magnetic ordering in this temperature range. Usually, heat capacity is mathematically expressed as $C = \gamma T + \beta T^3$, where the $\gamma T$ and $\beta T^3$ terms are relevant to the density of state and crystal lattice, respectively. The data at low temperatures below 7 K, shown in Fig. 5c, clearly manifests the existence of a linearly temperature-dependent term (the $\gamma$ term) despite the semiconducting ground state. The magnitude of $\gamma$ in 2–7 K is estimated at 12 ± 2 mJ K$^{-2}$ mol$^{-1}$ from the linear extrapolation of the $C/T$ versus $T^2$ plot down to zero K. This finite $\gamma$ value was reported in organic 2D triangular $\kappa$-(BEDT-TTF)$_2$Cu$_2$(CN)$_3$ and inorganic honeycomb H$_3$LiIr$_2$O$_6$ QSLs[32,46], suggesting possible QSL state of **1**. In addition, the black colour of the single-crystals with metallic

surfaces indicates that **1** could be electrically conductive. Figure 5d shows the temperature dependence of $\sigma$ in the range of 250–300 K, obtained with a two-probe dc method. The $\sigma$ value at 300 K is $(4.4–8.2) \times 10^{-5}$ S cm$^{-1}$ based on measuring several single-crystals. Parameter $\sigma$ decreases with temperature reduction, indicating that **1** behaves as a semiconductor. The activation energy ($E_a$) is determined to be 0.44 eV by using the Arrhenius function.

**Using generalised charge decomposition analysis (GCDA) method to understand donor–acceptor bonded interactions.** To further elucidate the electronic ground states, we first calculated the orbital energy for Cu$_2$(L)(py)$_4$ using the TD-DFT method. The calculated absorption spectrum coincides with the experimental data in the range of 300–1500 nm (Supporting Information Fig. 11 and Supporting Information Table 1). The strong absorption band at 920 nm (1.35 eV) is 100% attributed to ET from the HOMO (Fig. 6a) to LUMO (Fig. 6b). The electrons are predominantly located at both terminal Cu ions and the central ligand in the HOMO. Our calculations showed strong hybridisation with 58% $d_{xz}$(Cu) electron density and 25% $p_z$(N) electron densities, thus indicating that the electrons are delocalised in HOMO. Thus, the electron-intensive $d_{xz}$(Cu) orbitals are highly capable of donating electrons to the Li$^+$@C$_{60}$ cage through Cu–C bonds. The higher energy absorption band at 443 nm (2.80 eV) calculated at 402 nm (3.08 eV) is 78% assigned to ET from HOMO −1 to LUMO +1 (Supporting Information Fig. 12).

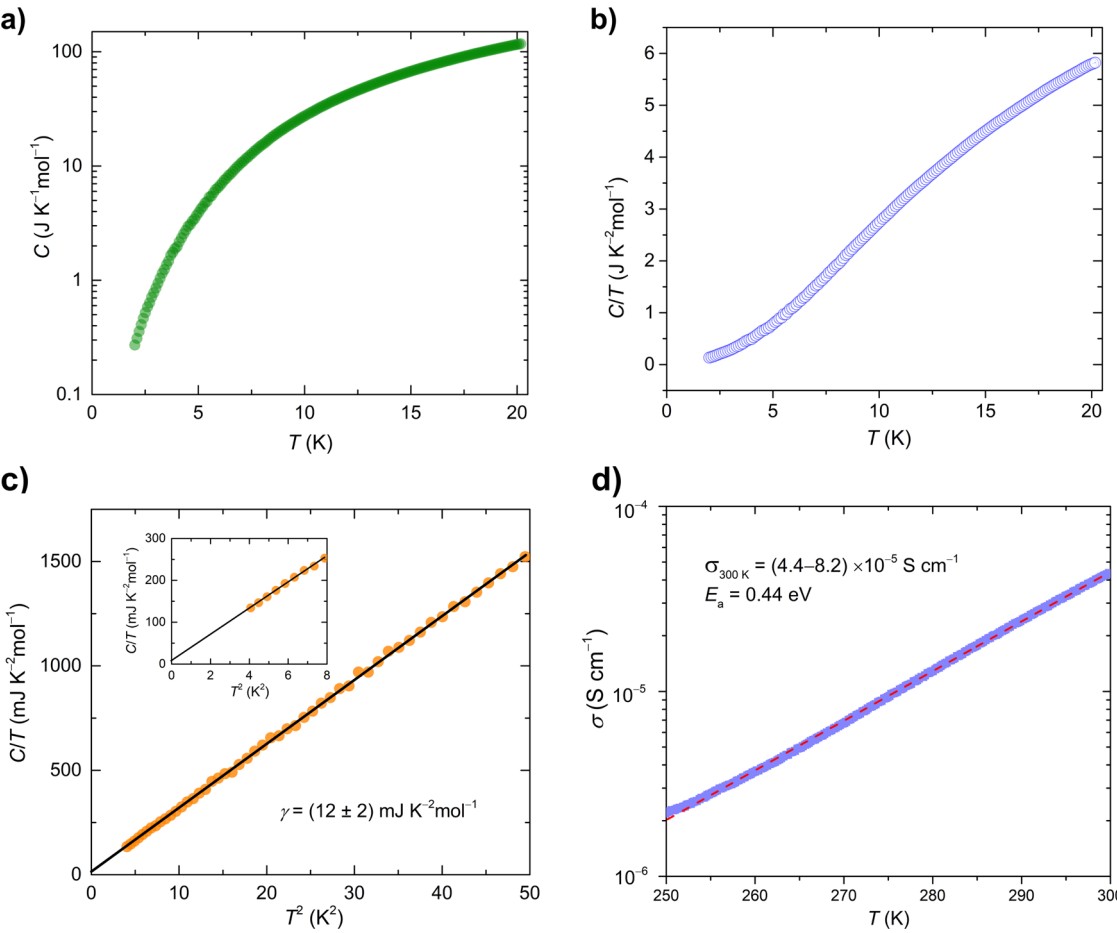

**Fig. 5 Heat capacity and temperature dependence of σ of 1. a** Temperature dependence of the total heat capacity ($C$) in the T range of 2.0–20 K in zero fields. **b** $C/T$ vs $T$ plot. **c** $C/T$ vs $T^2$ plot. **d** The Temperature dependence of $\sigma$ for single-crystals obtained by using a two-probe method in the range of 250–300 K.

To better understand the D–A bonding interactions, we used GCDA to investigate the electron-transfer amounts from $Cu_2L(py)_3$ (by adding a pyridine molecule to the $Cu_2L(py)_2$ molecule) to the Li$^+$@C$_{60}$ cage and the generation of complex orbitals by the fragments molecular orbitals in such an open-shell system. Figure 6c, d shows the calculated α and β forms of the orbital interaction diagrams for the triplet $Cu_2L(py)_3$Li$^+$@C$_{60}$ complex, respectively. The $[Cu_2L(py)_3]^+$ fragment was assumed to be a doublet with a positive charge, and the other fragment (Li$^+$@C$_{60}$$^{•−}$) contained an anionic radical in the C$_{60}$ cage (doublet). We calculated the net CT value as $0.275e^-$ (Supporting Information Table 2) by estimating the difference between electron donation and back-donation between the fragments. The partial CT interactions can be understood as follows: each $Cu_2(L)(py)_3$ can only transfer $0.25e^-$ to the Li$^+$@C$_{60}$ cage as the Li$^+$@C$_{60}$ cage is coordinated with four $Cu_2(L)(py)_2$. Thus, the GCDA calculation shows good agreement with the CV and magnetic results. From Fig. 6c, we note that the HOMO and LUMO of the complex are quite similar with an energy gap of 0.40 eV in the α form. The small bandgap is due to the significant contributions from HOMO of Li$^+$@C$_{60}$ and LUMO of $Cu_2(L)(py)_3$. The electron densities are mainly observed on the central ligand and the Li$^+$@C$_{60}$ cage. The HOMO of the complex originates from the mixture of 45% LUMO of $Cu_2L(py)_3$ and 32% HOMO and 21% LUMO of Li$^+$@C$_{60}$; similarly, the LUMO is mixed with 52% $Cu_2L(py)_3$ LUMO, 33% Li$^+$@C$_{60}$ HOMO, and 12% Li$^+$@C$_{60}$ LUMO. This result indicates that the electrons in the complex's HOMO are solely derived from the HOMO of Li$^+$@C$_{60}$;

there is no electron contribution from the occupied $Cu_2L(py)_3$ orbitals. Therefore, the complex's orbitals are not strongly mixed by fragments in the α form. In contrast, β electrons exhibit a deeper frontier orbital toward α electrons, indicating that β electrons are more electronically stable. Figure 6d shows that the frontier occupied orbitals of the complex in β form are doubly degenerated, indicating the significant contributions of fullerene orbitals. The electrons in the complex's HOMO are mainly delocalised on the Li$^+$@C$_{60}$ cage and a small fraction in the $d_{XZ}$ orbital of Cu ions, and these electrons originate from 11% HOMO, 4% HOMO−1 (Supporting Information Fig. 13), 2% HOMO−2 (Supporting Information Fig. 14) of $Cu_2L(py)_3$, 29% HOMO, 5% HOMO−1 (Supporting Information Fig. 15), and 42% HOMO−3 of Li$^+$@C$_{60}$. It is worth noting that a large number of delocalised $d_{XZ}$ electrons of Cu ions were observed in the HOMO of $Cu_2L(py)_3$; however, the number diminished in the complex's HOMO. This result strongly suggests that electrons transferred from $Cu_2L(py)_3$ to the Li$^+$@C$_{60}$ cage once Cu–C bonds were established. Thus, the bandgap is expected to be small because the orbital interactions between the fragments are strong. However, the calculated bandgap ($\Delta E = 1.17$ eV) is only slightly smaller than that of undoped Li$^+$@C$_{60}$ ($\Delta E \sim 1.5$ eV), which is probably caused by the strong onsite Coulomb interactions.

In summary, we succeeded in observing a conductive metal-fullerene-bonded framework with strong spin frustration, constructed by using redox-active $Cu_2(L)(py)_4$ and Li$^+$@C60 molecules. Via the precise control of the redox activities in each

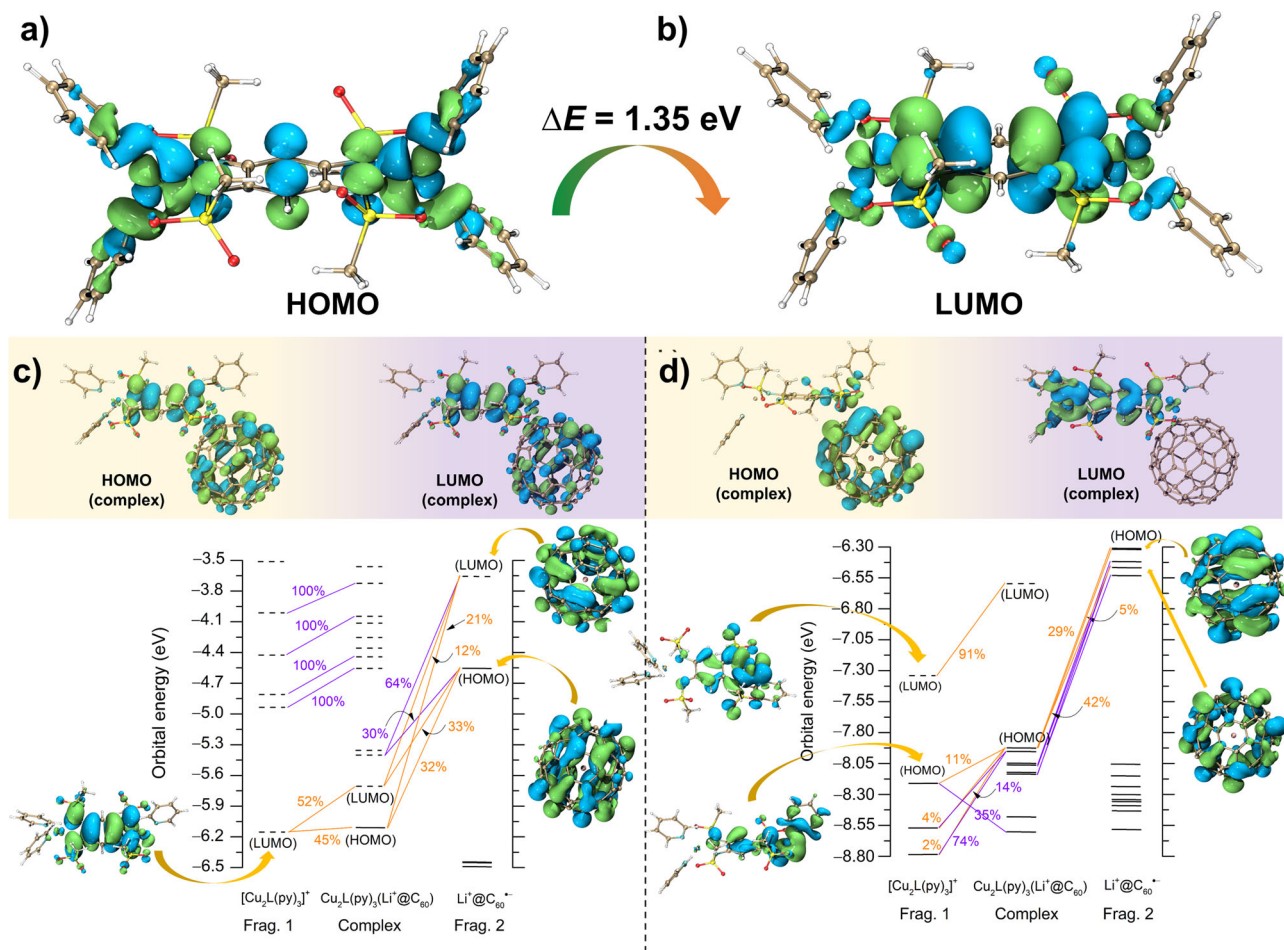

**Fig. 6 Orbital calculations and generalised charge decomposition analysis (GCDA) for donor-acceptor bonded interactions. a** HOMO of $Cu_2(L)(py)_4$. **b** LUMO of $Cu_2(L)(py)_4$. **c** Orbital interaction diagram and molecular orbitals in $\alpha$ electron form. **d** Orbital interaction diagram and molecular orbitals in $\beta$ electron form. The black solid lines and dotted lines represent the occupied and virtual orbitals, respectively. The orange and violet lines represent the contribution to the HOMO and LUMO and other orbitals, respectively. The isovalue for the electrons is set at 0.02.

species, the chemical bonds between the dinuclear electronic donor and the 3D spherical $Li^+@C_{60}$ acceptor allow for an interesting $S = \frac{1}{2}$ spin-lattice in a 1D ladder-like magnetic chain. The metal–fullerene bond accompanied by CT leads to a strong spin frustration ground state. Such a heterospin system is promising for the development of high-performance molecule-based spintronic devices and can aid in exploring new QSL candidates. Further studies in this direction aim to control the spin dynamics in a triangular or kagomé network using $Li^+@C_{60}$ superatoms.

## Methods

**Experimental synthesis**. A pure form of $(Li^+@C_{60})(NTf_2^-)$ was obtained from Idea International Co., Ltd. Sendai, Japan. $(Li^+@C_{60})(NTf_2^-)$ was synthesised by anion exchange from $(Li^+@C_{60})(PF_6^-)$. Typically, $Li^+@C_{60})(PF_6^-)$ (10 mg, 0.0115 mmol) and $LiNTf_2$ (5.0 mg, 0.0174 mmol) dissolved in 2.5 mL dichloromethane (DCM) and then sonicated 10 minutes to give a purple suspension. The suspension was filtered and the resulting clean solution was diffused by diethyl ether at 3 °C for 3 days to produce black solids. The solids were washed with a small amount of DCM and dried at room temperature (Yield: quantitative).

Synthesis of ligand ($H_4L$ = 1,2,4,5-tetrakis(methanesulfonamido)benzene). $H_4L$ was synthesised from the reaction of 1,2,4,5-tetraaminobenzene (138.2 mg, 1.0 mmol) and methylsulfonyl chloride (458.3 mg, 4.0 mmol) in 40 mL pyridine, and the resulting dark brown solution was stirred continuously for 3 h and then quenched in 15% aq. HCl. The resulting pale brown solid was collected by filtration, washed with distilled water (2 × 10 mL), and dried at 80 °C overnight (yield: 321 mg, 72%). IR in attenuated total reflection (ATR) mode: $\nu$(N–H) = 3250 cm$^{-1}$; $\nu$(CH$_3$) = 2933 cm$^{-1}$; $\nu$(S=O) = 1142 cm$^{-1}$.

Synthesis of $Cu_2(L)(py)_4$. $Cu(acetate)_2$ solid (363.3 mg, 2.0 mmol) was slowly added to 10 mL $H_4L$ (446.5 mg, 1.0 mmol) pyridine solution under dry $N_2$ gas. The resulting pale brown suspension immediately turned into a clear deep-blue solution, which was subsequently stirred for 12 h. Black block-like crystals were obtained by diethyl ether slowly diffused into the above solution (yield: 717.6 mg, 81%). The crystal structure was determined via single-crystal X-ray diffraction analysis. The purity was confirmed by CHN analysis (Supporting Information Table 3) and PXRD pattern (Supporting Information Fig. 16).

Synthesis of $\{[Cu_4(Li^+@C_{60})(L)(py)_4](NTf_2)(hexane)\}_n$ (**1**). This experiment was conducted in an argon-filled glovebox. First, $(Li^+@C_{60})(NTf_2^-)$ (1.0 mg, 0.001 mmol) was dissolved in 1 mL dry $o$-DCB, and the resulting pink solution was slowly added to $Cu_2(L)(py)_4$ (1.8 mg, 0.002 mmol) in 10.0 mL $o$-DCB solution. The molar ratio between $(Li^+@C_{60})(NTf_2^-)$ and $Cu_2(L)(py)_4$ was 1:2. The mixture solution immediately turned dark brown. It was stirred for 3 h, after which the solution was filtered. Small shiny black crystals with a typical size of 0.04 × 0.01 × 0.001 cm were obtained by slow diffusion with hexane in one week (yield: 1.0 mg, 33%). The purity was confirmed by CHN analysis (Supporting Information Table 2) and PXRD pattern (Supporting Information Fig. 17).

**Physical characterisation**. Single-crystal crystallographic data (exp_860) were collected at 120 K using a Rigaku Saturn 70 CCD diffractometer (Rigaku, Tokyo, Japan) with graphite monochromated Mo Kα radiation (λ = 0.71073 Å) generated by a VariMax microfocus X-ray rotating anode source. The single-crystal X-ray diffractions were measured at additional temperature points at 25, 50, 100, 200, and 300 K by using synchrotron radiation at Spring-8, Japan. The structures were solved by using Olex2 software[47]. Cyclic voltammetry (CV) was performed using an ALS/HCH Model 620D electrochemical analyser. A glassy carbon (3 mm diameter) electrode was used as the working electrode, a Pt wire was used as the counter electrode, and an Ag/Ag$^+$ was used as the reference electrode. The supporting electrolyte was 0.1 M tetrabutylammonium hexafluorophosphate (TBA·PF$_6$) in dry $o$-DCB. The solid-state absorption spectra were acquired by

compound **1** was embedded in KBr pellets with the use of a Shimadzu UV-3100PC instrument, and subsequently, the pellets were inserted into a transparent sealed cell and protected by argon gas. The solution-state absorption spectra were measured in a sealed plate tube filled with argon gas. The temperature dependence of the solid-state EPR spectra was examined by using the JEOL JES-FA100 device. Magnetic susceptibility measurements were conducted on a polycrystalline sample using a Quantum Design SQUID magnetometer (MPMS-7L). The magnetic susceptibility data were fitted by PHI software[48]. The temperature dependence of heat capacity was measured by using the Quantum Design PPMS. The temperature dependence of $\sigma$ was measured on single-crystals via a two-probe method by using the Quantum Design PPMS 6000 instrument.

**Quantum chemical calculation**. Geometry optimisation was performed by using density-functional theory (DFT) at the B3LYP/6-31G(d, p) level for pristine $Cu_2(L)(py)_4$. The absorption spectrum was calculated for the optimised geometry by using the TD-DFT method at the CAM-B3LYP/def2TZVP level. The $Cu_2(L)(py)_3(Li^+@C_{60})$ complex was cut from the X-ray structure by adding one pyridine molecule as the terminal ligand, and the resulting structure was optimised by applying DFT at the UB3LYP/6-31G(d, p) level. The optimised structure was decomposed into two fragments, $[Cu_2(L)(py)_3]^+$ and $Li^+@C_{60}^{\bullet-}$, for single-point calculation. The GCDA[49] was used for the open-shell form of this complex and the two fragments by applying DFT at the UB3LYP/def2tzvp level. All calculations were performed using Gaussian 16 software[50], and the output results were analysed using the Multiwfn programme[49,51].

## Data availability

The crystallographic data generated in this study have been deposited in the CCDC database under accession codes 2039740 (compound **1** at 120 K), 2039286 ($Cu_2(L)(py)_4$ at 120 K), and 2121505–2121509 (compound **1** at 25, 50, 100, 200, and 300 K). The data generated in this manuscript are available within the manuscript, its supplementary information, or from the authors upon request.

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

## Acknowledgements

This work was supported by JSPS KAKENHI Grant Numbers JP19H05631 (S.T. and M.Y.). M.Y. acknowledges the support of the 111 Project (B18030) from China. Computations were carried out using the computer resources offered under the category of the Trial Use Project by the Research Institute for Information Technology, Kyushu University.

## Author contributions

Y.S. conceived and designed the project. Y.S. synthesised, characterised, and analysed all compounds. M.C. measured the optical spectra. K.S. measured low-temperature single-crystal X-ray diffraction. E.K., K.K. and Y.K. provided $(Li^+@C_{60})(NTf_2)$ salt. Y.S., T.Y., N.H., and T.A. measured the EPR spectra. H.K and T.F. measured heat capacity. Y.S. performed and analysed the GCDA and DT-DFT calculations. Y.S. wrote the paper with input from M.Y., S.T., T.T.H.K. and A.O.

## Competing interests

The authors declare no competing interests.
