## [Peer Review File · Nature Communications]

Heterospin Frustration in a Metal–Fullerene–Bonded Semiconductive AntiferromagnetREVIEWER COMMENTS

Reviewer #1 (Remarks to the Author):

In my opinion, the work is interesting and well founded. The results are remarkable and constitute an important contribution to the field. Furthermore, the conclusions are duly supported by the results.

However, I have a couple of objections. The first one is not very important, but the second one I think it is mandatory that it be corrected:

1) In the Introduction section I miss some important references of calculations made on Computational studies about encapsulated lithium cation in C60. Without wishing to be exhaustive, some of them of relevance:

V. Bernshtein, I. Oref, Phys. Rev. A, 2000, 62, 03320.

H. U. Rehman, N. A. McKee, M. L. McKee, J. Comput.Chem., 2016, 37, 194.

I. Gonzalez-Veloso, J. Rodriguez-Otero, E. M. Cabaleiro-Lago, PCCP; 2019, 21, 16665.

2) The choice of the method for obtaining the absorption spectrum (TD-DFT method at the B3LYP/6-311G(d,p) level) is not the best possible one. It is widely established that to obtain an acceptable reproduction of the absorption spectrum, the use of a long-range corrected functional, such as LC- ω PBE or CAM-B3LYP, is highly recommended.

Therefore, it would be mandatory that the authors repeat the quantum mechanical calculations with a more appropriate functional. Perhaps the results are not very different, but I think that the method recommended by the literature should always be tried.

Reviewer #2 (Remarks to the Author):

The manuscript by Shen and coworkers details the structure and physical properties of a new material assembled from Li@C60 and Cu complexes. The synthetic and structural results are impressive but the authors make claims about the magnetic properties of this materials that are not supported by the data presented in the manuscript. For this reason, I cannot recommend publication in the current form. If the authors can bulk up the magnetism with more measurement and/or analysis section to support their claim of spin liquid, or re-write the paper to shift the focus, then I could reconsider my recommendation.

Here are a few things to consider.

1. The fit of $1/X$ data to extract the Weiss temperature is not valid as there is no truly linear region. The authors should subtract a temperature independent contribution. Take a look at this paper for an in-depth discussion: <https://www.sciencedirect.com/science/article/pii/S0304885316324581>
2. Spin liquid are notoriously difficult to probe. The authors should look into other methods to show there is still quantum fluctuations at low T, including muon spin relaxation and ac susceptibility.
3. Figure 2 is useless. Figure is also difficult to visualize the structure. I find the SI Figures are actually better than those in manuscript.
4. There is no discussion of the synthesis in the manuscript.
5. Page 3, Line 58: The authors claim that it is easy to deduce that Li@C60 can be doped with alkali metals to reach a superconducting state. I am not aware of any reports of superconductivity in Li@C60. I assume the authors meant something else. The text should be modified.
6. The authors talk about bandgap on line 169. It's not clear to me that these materials are actually band semiconductors. The authors should be careful with this.
7. I'm still not convinced the structure is geometrically frustrated. It relies on a number of hypothesis. Is the structure distorting at low T, which would remove the frustration? Is the radical localized at certain T?

Reviewer #3 (Remarks to the Author):

This work reports the synthesis and characterization of an impressive 1D-coordination polymer, constructed upon the reaction of the dinuclear copper(II) complex $[\text{Cu}_2\text{L}(\text{py})_4]$ ($\text{L} = 1,2,4,5$ -tetrakis(methanesulfonamido)benzene) with the lithium encapsulated fullerene salt $(\text{Li}^+ @ \text{C}_{60})(\text{NTf}_2^-)$ ($\text{NTf}_2^- = \text{bis}(\text{trifluoromethane})\text{sulfonamide anion}$), forming compound $\{[\text{Cu}_4(\text{Li}^+ @ \text{C}_{60})\text{L}(\text{py})_4](\text{NTf}_2)(\text{hexane})\}_n$ (1). Compound 1 describes a 1D-coordination polymer (chain) in which each $(\text{Li}^+ @ \text{C}_{60})$ “molecule” coordinates to four Cu(II) centres belonging to two $[\text{Cu}_2\text{L}(\text{py})_4]$, while the remaining Cu(II) centres serve as nodes to a neighboring $(\text{Li}^+ @ \text{C}_{60})$ “molecule”, thus resulting in the chain-like motif. Electrical conductivity studies reveal that 1 displays long-range electrical conductivity, while magnetic studies demonstrate that 1 may be treated as a spin-frustrated system.

The quality of the ms. is good, although at some points the authors should revise the use of the English language. All data presented fully agree with the analysis presented by the authors, while in addition the theoretical studies performed support their claims.

My personal view is that this work is quite novel and exciting, since it is the first time that such exotic coordination polymers exhibit promising electrical and magnetic properties. I believe the results reported in this work will be of great significance to the fields of inorganic/coordination chemistry, magnetochemistry, physics and materials, with potential applications in spintronic devices. Therefore, I am happy to suggest acceptance of the ms. in Nature Communications, since it will attract the wider readership of scientists working in the above-mentioned fields.

The points the authors should consider are the following ones:

- 1) Regarding the purity of the bulk samples included in the work (metallic precursor and compound 1) no data are presented besides the single-crystal structure. Therefore, for each compound: a) elemental C,H,N analysis should be provided, b) p-XRD diagrams should be provided along with comparison with the theoretical patterns, c) EDS measurements should be provided regarding the metallic content of compound 1.
- 2) In 1 the coordination environment of the Cu centres is described as distorted trigonal bipyramidal. The distortion should be quantified in terms of the deviation parameter.
- 3) No deviation is given in the magnetic exchange parameters and the goodness value of the fit is missing.
- 4) I can understand the existence of five exchange parameters for the fitting of the magnetic susceptibility, since there are indeed five different pathways. However, so many parameters often lead to overparameterization or to conclusions with no physical/real meaning. I would suggest the authors to perform the analysis with four J exchange parameters (i.e. $J_1\text{-radical} = J_2\text{-radical}$) and report/compare the two different models (in the SI).

Reviewer #4 (Remarks to the Author):

This manuscript reports a novel coordination polymer constructed by using $\text{Li}^+ @ \text{C}_{60}$ as acceptor and a specific Cu complex as donor. In the obtained framework, each $\text{Li}^+ @ \text{C}_{60}$ coordinates with four Cu^{2+} ions forming infinite 1D ladder-like patterns along the crystallographic b-axis. This is accompanied by the strong charge transfer from the Cu species to the fullerene core. As a result, the four Cu^{2+} ions ($S = 1/2$) and $\text{Li}^+ @ \text{C}_{60}$ ($S = 1/2$) interact with each other, showing magnetic frustration in a triangular-like lattice.

The paper is well-presented, the experiments and theoretical analysis are very solid. Although some fullerene-based 1D polymers have been reported previously, the assembly of $\text{Li}^+ @ \text{C}_{60}$ with a Cu complex is a new finding. Moreover, the obtained coordination polymer is conductive and features strong spin frustration. I would recommend the acceptance of this paper after addressing some minor issues.

1. Is the electrical conductivity of the crystal anisotropic? For example, along the ladder-chain direction (Figure 1d) versus other directions.
2. The Z value in the crystal data of 1 (Datablock: exp_860) should be corrected. Although the 1D ladder-like structure of this crystal is clear, I would still suggest the authors to do more refinement on

the anion NTf₂⁻ moieties and the disordered hexane molecules.

3. Some typos:

a. In page 2, line 28, the close brace is missing;

b. Please unify/check the writing of ions/compounds. For example, NTf₂ versus NTf₂⁻;

'[Li+@C60](SbCl₆)' in Page 3, line 45.

Response to reviewers' comments

We thank the four reviewers' great efforts in reviewing our manuscript. All modifications in the revised manuscript are highlighted with a yellow background. A point-by-point response to the reviewers' comments are shown below (reviewers' comments in blue, authors response in black):

Reviewer #1 (Remarks to the Author):

In my opinion, the work is interesting and well founded. The results are remarkable and constitute an important contribution to the field. Furthermore, the conclusions are duly supported by the results.

However, I have a couple of objections. The first one is not very important, but the second one I think it is mandatory that it be corrected:

Authors response: Thank you so much for reviewing our manuscript and positive comments.

1) In the Introduction section I miss some important references of calculations made on Computational studies about encapsulated lithium cation in C₆₀. Without wishing to be exhaustive, some of them of relevance:

V. Bernshtein, I. Oref, Phys. Rev. A, 2000, 62, 03320.

H. U. Rehman, N. A. McKee, M. L. McKee, J. Comput.Chem., 2016, 37, 194.

I. Gonzalez-Veloso, J. Rodriguez-Otero, E. M. Cabaleiro-Lago, PCCP; 2019, 21, 16665.

Authors response: After carefully checking these references, we added these references in the revised manuscript (in ref 4, 6, 11) because they helped Nature Communications readers to better understand the C₆₀ electronic structures.

2) The choice of the method for obtaining the absorption spectrum (TD-DFT method at the B3LYP/6-311G(d,p) level) is not the best possible one. It is widely established that to obtain an acceptable reproduction of the absorption spectrum, the use of a long-range corrected functional, such as LC- ω PBE or CAM-B3LYP, is highly recommended.

Therefore, it would be mandatory that the authors repeat the quantum mechanical calculations with a more appropriate functional. Perhaps the results are not very different, but I think that the method recommended by the literature should always be tried.

Authors response: In the revised manuscript, we tried the TD-DFT calculation by using CAM-B3LYP/de2tzvp and B3LYP/def2tzvp for Cu₂(L)(py)₄. The electron transitions from HOMO to LUMO were observed at 920, 765, 760, 860 nm by using experimental, B3LYP/6-311G(d,p), B3LYP/def2tzvp and CAM-B3LYP/def2tvp, respectively. The results are summarized in the table and figures. From the results, CAM-B3LYP/de2tzvp method showed better results.

Basis sets	band (1)	band (2)
Experimental	920 nm	444 nm
B3LYP/6-311G(d,p)	765 nm	412
B3LYP/def2tzvp	760 nm	340 nm
CAM-B3LYP/def2tzvp	860 nm	470 nm, 507 nm

Reviewer #2 (Remarks to the Author):

The manuscript by Shen and coworkers details the structure and physical properties of a new material assembled from Li@C60 and Cu complexes. The synthetic and structural results are impressive but the authors make claims about the magnetic properties of this materials that are not supported by the data presented in the manuscript. For this reason, I cannot recommend publication in the current form. If the authors can bulk up the magnetism with more measurement and/or analysis section to support their claim of spin liquid, or re-write the paper to shift the focus, then I could reconsider my recommendation.

Authors response: Thank you so much for reviewing our manuscript and comments.

Here are a few things to consider.

1. The fit of $1/X$ data to extract the Weiss temperature is not valid as there is no truly linear region. The authors should subtract a temperature independent contribution.

Take a look at this paper for an in-depth

discussion: <https://www.sciencedirect.com/science/article/pii/S0304885316324581>

Authors response: Thank you so much for your reference. Based on this reference, we subtracted the temperature independence paramagnetism (χ_0), the χ_0 value was estimated to be $8.8 \times 10^{-4} \text{ cm}^3 \text{ mol}^{-1}$ in 1 T field by extrapolating $(\chi - \chi_0)$ vs T^{-1} curve to $T^{-1} \rightarrow 0$. In addition, we examined the effects on $\chi - \chi_0$ by changing χ_0 values from 0 to $5 \times 10^{-3} \text{ cm}^3 \text{ mol}^{-1}$. The Weiss temperatures (θ_{cw}) were varied from -305 to 25 K. In our case, the θ_{cw} value was estimated to be -190 K when $\chi_0 = 8.8 \times 10^{-4} \text{ cm}^3 \text{ mol}^{-1}$.

2. Spin liquid are notoriously difficult to probe. The authors should look into other methods to show there is still quantum fluctuations at low T, including muon spin relaxation and ac susceptibility.

Authors response: To probe spin liquid, muon spin relaxation is powerful, however, we don't have such collaborators who can measure in the UK. Alternatively, we think heat capacity is another powerful tool to unearth hidden magnetic phases and has been widely used for the spin liquid system. Therefore, we performed low-temperature heat capacity and ac susceptibility measurements. Both the results indicated there is no anomaly in 2-20 K, suggesting no magnetic ordering in this temperature range.

Usually, heat capacity is given by γT (density of state) + βT^3 (lattice). The γ is finite in metals, but it is 0 in insulators. Interestingly, the γ value of compound **1** is estimated to be (12 ± 2) mJ K⁻² mol⁻¹ in an semiconducting ground state ($T < 7$ K) and is quite similar to that of quantum spin liquids candidates in organic and inorganic crystals as reported by the following two papers: 1) Kitagawa, K., Takayama, T., Matsumoto, Y. *et al.* A spin-orbital-entangled quantum liquid on a honeycomb lattice. *Nature* **554**, 341–345 (2018). 2) Yamashita, S., Nakazawa, Y., Oguni, M. *et al.* Thermodynamic properties of a spin-1/2 spin-liquid state in a κ -type organic salt. *Nature Phys* **4**, 459–462 (2008). These results indicated that compound **1** is a possible candidate of QSL.

We added these results to the revised manuscript to give more evidence of spin liquid of **1**.

3. Figure 2 is useless. Figure is also difficult to visualize the structure. I find the SI Figures are actually better than those in manuscript.

Authors response: Thank you so much for your suggestions. We moved Figure 2 to the Supporting Information and Figure S1 moved to the manuscript. In addition, we rearranged the figures in the revised manuscript.

4. There is no discussion of the synthesis in the manuscript.

Authors response: We added short discussion about the synthetic procedure in the revised manuscript.

5. Page 3, Line 58: The authors claim that it is easy to deduce that Li@C60 can be doped with alkali metals to reach a superconducting state. I am not aware of any reports of superconductivity in Li@C60. I assume the authors meant something else. The text should be modified.

Authors response: We are sorry for this confusing expression. We agree that there is no superconductivity reported so far about Li@C60. In this text, we would like to express the potential presence of superconductivity in Li@C60 by comparison of A3C60 superconductors. We have revised this text as ‘It is possible that Li⁺@C₆₀ can be doped by alkali metals to produce A₃(Li@C₆₀) (A = K⁺, Rb⁺ and Cs⁺) species in imitation of M₃C₆₀ superconductors with three electrons accommodated in the triply degenerated LUMO.’

6. The authors talk about bandgap on line 169. It’s not clear to me that these materials are actually band semiconductors. The authors should be careful with this.

Authors response: We revised this sentence in the revised manuscript.

7. I’m still not convinced the structure is geometrically frustrated. It relies on a number of hypothesis. Is the structure distorting at low T, which would remove the frustration? Is the radical localized at certain T?

Authors response: Thanks for your good question. In the revised manuscript, we determined the low-temperature single-crystal structures by synchrotron radiation measurements CCDC NO: 2121505-2121509. From the structure and PXRD analysis, first, we did not observe the structural changes in 25–300 K; second, Li⁺ ion inside the C₆₀ cage was found to be split into half and localized at a certain position from 100 K to 25 K. The position of Li⁺ ion remained unchanged below 100 K. Above 100

K, we did not detect the Li^+ position due to the fast motion of the Li^+ ion. As the localized Li^+ ion would strongly attract the negative radical (due to Li^+-C bonds), the radicals should localize on the six-carbon ring (pink carbon atoms in Figure 2).

Reviewer #3 (Remarks to the Author):

This work reports the synthesis and characterization of an impressive 1D-coordination polymer, constructed upon the reaction of the dinuclear copper(II) complex $[\text{Cu}_2\text{L}(\text{py})_4]$ ($\text{L} = 1,2,4,5$ -tetrakis(methanesulfonamido)benzene) with the lithium encapsulated fullerene salt $(\text{Li}^+@C_{60})(\text{NTf}_2^-)$ ($\text{NTf}_2^- =$ bis(trifluoromethane)sulfonamide anion), forming compound $\{[\text{Cu}_4(\text{Li}^+@C_{60})\text{L}(\text{py})_4](\text{NTf}_2)(\text{hexane})\}_n$ (1). Compound 1 describes a 1D-coordination polymer (chain) in which each $(\text{Li}^+@C_{60})$ “molecule” coordinates to four Cu(II) centres belonging to two $[\text{Cu}_2\text{L}(\text{py})_4]$, while the remaining Cu(II) centres serve as nodes to a neighboring $(\text{Li}^+@C_{60})$ “molecule”, thus resulting in the chain-like motif. Electrical conductivity studies reveal that 1 displays long-range electrical conductivity, while magnetic studies demonstrate that 1 may be treated as a spin-frustrated system.

The quality of the ms. is good, although at some points the authors should revise the use of the English language. All data presented fully agree with the analysis presented by the authors, while in addition the theoretical studies performed support their claims.

My personal view is that this work is quite novel and exciting, since it is the first time that such exotic coordination polymers exhibit promising electrical and magnetic properties. I believe the results reported in this work will be of great significance to the fields of inorganic/coordination chemistry, magnetochemistry, physics and materials, with potential applications in spintronic devices. Therefore, I am happy to suggest acceptance of the ms. in Nature Communications, since it will attract the wider readership of scientists working in the above-mentioned fields.

Authors response: Thank you so much for your positive comments.

The points the authors should consider are the following ones:

1) Regarding the purity of the bulk samples included in the work (metallic precursor and compound 1) no data are presented besides the single-crystal structure. Therefore, for each compound: a) elemental C,H,N analysis should be provided, b) p-XRD diagrams should be provided along with comparison with the theoretical patterns, c) EDS measurements should be provided regarding the metallic content of compound 1. Authors response: The elemental (CHN) analysis was performed in the *Research and Analytical Center for Giant Molecules* at Tohoku University; the results are tabulated as below:

Compounds	C (%)	H (%)	N (%)
	Cal. / Exp.	Cal. / Exp.	Cal. / Exp.
$\text{Cu}_2(\text{L})(\text{py})_4$	40.49 / 40.46	3.85 / 3.86	12.59 / 12.47
$\{[\text{Cu}_4(\text{Li}^+@C_{60})\text{L}(\text{py})_4](\text{NTf}_2)(\text{hexane})\}_n$	55.76 / 55.34	2.29 / 1.96	5.97 / 5.71

The PXRD patterns for $\text{Cu}_2(\text{L})(\text{py})_4$ (upper) and compound 1 (down).

2) In 1 the coordination environment of the Cu centres is described as distorted trigonal bipyramidal. The distortion should be quantified in terms of the deviation

parameter.

Authors response: Thank you so much for your comments. Unlike the PF₅ and PCl₅, they have a D_{3h} symmetry, Cu(N)₃(C)₂ has a distorted trigonal bipyramidal geometry because of the spherical surface of C₆₀ and the two carbon atoms are bonded.

Therefore, it shows very low symmetry and lacks of axis. Compared to trigonal bipyramidal geometry with a D_{3h} symmetry, the deviation parameters are defined as (angle-90°)/90° and (180°-angle)/180° and calculated to be 15.6%, 21.1% and 22.2%.

Figure: Cu(N)₃(C)₂ (left)

PF₅, D_{3h} symmetry(right)

$$\begin{aligned} \angle \text{N5Cu1N2} &= 104.1^\circ & \angle \text{N5Cu1N1} &= 104.1^\circ \\ \angle \text{N5Cu1C1} &= 109.0^\circ & \angle \text{N5Cu1C6} &= 141.9^\circ \end{aligned}$$

So, deviation parameters P are:

$$P1 = (104.1-90) / 90 * 100\% = 15.6\%$$

$$P2 = (109-90) / 90 * 100\% = 21.1\%$$

$$P3 = (180-140) / 180 * 100\% = 22.2\%$$

3) No deviation is given in the magnetic exchange parameters and the goodness value of the fit is missing.

Authors response: The effective exchange coupling parameters of $J_{\text{Cu-Cu}} = -170 \pm 5$ K and $J_{1-\text{C}_{60}} = J_{2-\text{C}_{60}} = -185 \pm 3$ K with $g_{\text{Cu}} = 2.09(0)$ were obtained after the best fit ($g_{\text{C}_{60}}$ was fixed to 2.0). The R -value is 1.9×10^{-3} .

4) I can understand the existence of five exchange parameters for the fitting of the magnetic susceptibility, since there are indeed five different pathways. However, so many parameters often lead to overparameterization or to conclusions with no physical/real meaning. I would suggest the authors to perform the analysis with four J exchange parameters (i.e. J1-radical = J2-radical) and report/compare the two different models (in the SI).

Authors response: We reduced the parameters by only considering the exchange coupling $J_{\text{Cu-Cu}}$ and $J_{\text{Cu-C}_{60}}$. To obtain the J values, we used the following spin Hamiltonian \hat{H} using equations (1) by considering two kinds of exchange coupling $J_{\text{Cu-Cu}}$ and $J_{\text{Cu-C}_{60}}$.

$$\begin{aligned} \hat{H} = & -2J_{\text{Cu-Cu}}(\hat{S}_{\text{Cu1}} \cdot \hat{S}_{\text{Cu2}} + \hat{S}_{\text{Cu3}} \cdot \hat{S}_{\text{Cu4}}) - 2J_{\text{Cu-C}_{60}}(\hat{S}_{\text{Cu1}} \cdot \hat{S}_{\text{C}_{60}^+} + \hat{S}_{\text{Cu2}} \cdot \hat{S}_{\text{C}_{60}^+} + \hat{S}_{\text{Cu3}} \cdot \hat{S}_{\text{C}_{60}^-} + \hat{S}_{\text{Cu4}} \cdot \hat{S}_{\text{C}_{60}^-}) \\ & + \mu_B \vec{B} \cdot g_{\text{Cu}} \cdot \hat{S}_{\text{Cu}} + \mu_B g_{\text{C}_{60}} \vec{B} \cdot \hat{S}_{\text{C}_{60}} \end{aligned} \quad (1)$$

To avoid overparameterization, we assumed the interactions of J_{14} and J_{23} are ignored compared to J_{12} , J_{34} and $J_{\text{Cu-C}_{60}^-}$ due to their long metal-metal distances and the interactions between C_{60}^- and Cu1, Cu2 are equivalent. The effective exchange coupling parameters of $J_{\text{Cu-Cu}} = -170 \pm 5$ K and $J_{1-\text{C}_{60}^-} = J_{2-\text{C}_{60}^-} = -185 \pm 3$ K with $g_{\text{Cu}} = 2.09(0)$ were obtained after the best fit ($g_{\text{C}_{60}^-}$ was fixed to 2.0).

Reviewer #4 (Remarks to the Author):

This manuscript reports a novel coordination polymer constructed by using $\text{Li}^+@C_{60}$ as acceptor and a specific Cu complex as donor. In the obtained framework, each $\text{Li}^+@C_{60}$ coordinates with four Cu^{2+} ions forming infinite 1D ladder-like patterns along the crystallographic b-axis. This is accompanied by the strong charge transfer from the Cu species to the fullerene core. As a result, the four Cu^{2+} ions ($S = 1/2$) and $\text{Li}^+@C_{60}^-$ ($S = 1/2$) interact with each other, showing magnetic frustration in a triangular-like lattice.

The paper is well-presented, the experiments and theoretical analysis are very solid. Although some fullerene-based 1D polymers have been reported previously, the assembly of $\text{Li}^+@C_{60}$ with a Cu complex is a new finding. Moreover, the obtained coordination polymer is conductive and features strong spin frustration. I would recommend the acceptance of this paper after addressing some minor issues.

1. Is the electrical conductivity of the crystal anisotropic? For example, along the ladder-chain direction (Figure 1d) versus other directions.

Authors response: Thank you so much for your positive comments. Unfortunately, the single-crystals morphology is not so nice and the crystal size is relatively small. We cannot determine the a , b , c axes by using such a single crystal. However, we can expect the conductive pathway from the crystal structure. One conductive pathway could be the 1D ladder-like chain as electrons move from the donor $\text{Cu}_2(\text{L})(\text{py})_2$ to the acceptor $\text{Li}^+@C_{60}$ cage (through bond); the other pathway is the electrons move between $\text{Li}^+@C_{60}$ cages along the a -axis (pi-pi stacking, through space).

2. The Z value in the crystal data of 1 (Datablock: exp_860) should be corrected. Although the 1D ladder-like structure of this crystal is clear, I would still suggest the authors to do more refinement on the anion NTf_2^- moieties and the disordered hexane molecules.

Authors response: We resolved the crystal structures and reduced and explained the alert A level in the cif file.

3. Some typos:

a. In page 2, line 28, the close brace is missing;

Authors response: We added it in the revised manuscript.

b. Please unify/check the writing of ions/compounds. For example, NTf_2^- versus

NTf2-; '[Li+@C60](SbCl6)' in Page 3, line 45.

Authors response: We corrected them in the revised manuscript.

REVIEWER COMMENTS

Reviewer #1 (Remarks to the Author):

All my objections were fully addressed and therefore I think the paper is suitable for publication. Only a small mistake: in the legend of figure S11 the experimental data are not shown in orange but in black

Reviewer #2 (Remarks to the Author):

The authors made a valiant effort to answer my comments. Considering how challenging it is to characterize spin liquids and how careful the authors are in the revised manuscript, I recommend accepting this work for publication.

Reviewer #3 (Remarks to the Author):

In the revised ms., the authors answered with clarity to all issues initially raised. Therefore, I am more than happy to suggest acceptance of the manuscript at its current form.

Reviewer #4 (Remarks to the Author):

I think the authors have fully addressed the concerns from all four reviewers. The texts and experimental data are very compelling. In my opinion it is now ready to be published without further change.

Response to reviewers' comments (2nd round)

We thank the four reviewers again for reviewing our revised manuscript. A point-by-point response to the reviewers' comments are shown below (reviewers' comments in blue, authors response in black):

Reviewer #1 (Remarks to the Author):

All my objections were fully addressed and therefore I think the paper is suitable for publication.

Only a small mistake: in the legend of figure S11 the experimental data are not shown in orange but in black.

Authors response: Thank you so much for pointing it out. We have corrected them in the supporting information.

Reviewer #2 (Remarks to the Author):

The authors made a valiant effort to answer my comments. Considering how challenging it is to characterize spin liquids and how careful the authors are in the revised manuscript, I recommend accepting this work for publication.

Authors response: Thank you so much for your comments. We tried our best to probe the possible presence of a quantum spin liquid state in compound **1**.

Reviewer #3 (Remarks to the Author):

In the revised ms., the authors answered with clarity to all issues initially raised. Therefore, I am more than happy to suggest acceptance of the manuscript at its current form.

Authors response: Thank you so much for your acceptance.

Reviewer #4 (Remarks to the Author):

I think the authors have fully addressed the concerns from all four reviewers. The texts and experimental data are very compelling. In my opinion it is now ready to be published without further change.

Authors response: Thank you so much for your recommendation.